# Genome-Wide Identification and Analysis of Lbd Transcription Factor Genes in *Jatropha curcas* and Related Species

**DOI:** 10.3390/plants11182397

**Published:** 2022-09-14

**Authors:** Qi Jin, Zitian Yang, Wenjing Yang, Xiaoyang Gao, Changning Liu

**Affiliations:** 1School of Life Sciences, University of Science and Technology of China, Hefei 230026, China; 2CAS Key Laboratory of Tropical Plant Resources and Sustainable Use, Xishuangbanna Tropical Botanical Garden, Chinese Academy of Sciences, Kunming 650223, China

**Keywords:** *LBD*, *Jatropha curcas*, gene family, phylogenetic analysis

## Abstract

Lateral organ boundaries domain (LBD) proteins are plant-specific transcription factors that play important roles in organ development and stress response. However, the function of *LBD* genes has not been reported in Euphorbiaceae. In this paper, we used *Jatropha curcas* as the main study object and added rubber tree (*Hevea brasiliensis*), cassava (*Manihot esculenta* Crantz) and castor (*Ricinus communis* L.) to take a phylogenetic analysis of *LBD* genes. Of LBD, 33, 58, 54 and 30 members were identified in *J. curcas*, rubber tree, cassava and castor, respectively. The phylogenetic analysis showed that *LBD* members of Euphorbiaceae could be classified into two major classes and seven subclasses (Ia-Ie,IIa-IIb), and *LBD* genes of Euphorbiaceae tended to cluster in the same branch. Further analysis showed that the *LBD* genes of Euphorbiaceae in the same clade usually had similar protein motifs and gene structures, and tissue expression patterns showed that they also have similar expression profiles. *JcLBD*s in class Ia and Ie are mainly expressed in male and female flowers, and there are multiple duplication genes with similar expression profiles in these clades. It was speculated that they are likely to play important regulatory roles in flower development. Our study provided a solid foundation for further investigation of the role of *LBD* genes in the sexual differentiaion of *J. curcas*.

## 1. Introduction

Transcription factors (TFs) can play significant roles in plant development by influencing gene expression. The *LBD* gene family, also known as the *asymmetric leaves2-like* (*ASL*) gene family, is a group of plant-specific transcription factors that encode proteins containing lateral organ boundaries (LOB) domains [1,2]. Previous studies have shown that *LBD* genes were predominantly expressed in cells located on the base of the adaxial axis in all lateral organs formed by the shoot apical meristem, as well as at the base of the lateral roots. Overexpression of *AtLOB/AtASL4* results in a smaller plant type and changes in the shape and size of flower organs, implying that *LBD* genes play important roles in the development of lateral organs [3]. The *LBD* genes can be split into two major classes and seven subclasses (Ia-e, IIa and IIb) based on sequence similarity and phylogenetic trees. The N-terminal of each *LBD* gene is largely conserved, and the C-terminal is varied. The LOB domain is at the N-terminal end of the protein, which contains a conserved CX2CX6CX3C zinc finger-like motif. Class I, which most *LBD* genes belong to, contains a glycine-alanine-serine (GAS-block) region and a LX6LX3LX6L leucine zipper-like helical coiled-coil structure behind the CX2CX6CX3C zinc finger region [1,4,5].

Previous studies have identified that *LBD* genes participate in the development of many lateral organs [3,4]. For example, the *AtLOB* gene can regulate the development of young leaves [3]; homologs of *AtLOB* have been identified in other species: *Romaso2* in maize regulates the development of ears [6], *OsRA2* in rice is involved in modifying panicle architecture through regulating pedicel length [7], *HvRA2* (*Vrs4*) in barley can regulate lateral spikelet fertility, the *vrs4* mutant displays a loss of spikelet determinacy [8]; *AtLBD10* and *AtLBD27* can regulate pollen development and all pollen grains aborted in the *lbd10 lbd27* double mutant [9]. Recent studies have revealed that in addition to their role in lateral organs, they are also involved in many other biological processes: *MdLBD13* in apple can inhibit anthocyanin synthesis and nitrogen utilization via the flavonoid pathway [10]; *EgLBD29* and *EgLBD37* in *Eucalyptus grandis* can affect phloem fibre production and secondary xylem, respectively [11]; *SlLBD40* is a negative regulator of drought tolerance, and *sllbd40* knockout mutants with higher drought tolerance than widetype tomato [12]. Based on the functional reports of AtLBD proteins, it was found that *LBD* genes from the same phylogenetic branch tend to have similar functions: Class Ia proteins may regulate aboveground organs development [3,9,13]; Class Ib proteins mainly regulate lateral root formation [14,15]; and Class II LBD proteins play an important function in nitrogen response and anthocyanin synthesis [16,17].

The *LBD* gene family has been studied in several species, but so far there is no report about the function of *LBD* genes in Euphorbiaceae. *Jatropha curcas* is an important cash crop in Euphorbiaceae. All the parts of the *J. curcas* plant have important economic value, especially its seeds. The seeds of *J. curcas* are considered as important material for biodiesel production because they contain about 40% oil and do not produce harmful substances after burning [18]. However, the yield of *J. curcas* seeds is low, which is less than 1 ton per hectare a year under normal growth conditions. This limits the application of *J. curcas* as a biodiesel. *J. curcas* is a dioecious plant, and the ratio of female to male flowers is low, generally 1:29-1:13. The low ratio of female to male of *J. curcas* is an improtant factor that leads to a low yield of seeds. *J. curcas* have two sex determination patterns: male flowers are unisexual from early development; female flowers are bisexual first, and after the sixth stage of development, stamens appear abortive and eventually produce mature fertile female flowers [19,20,21]. Rubber tree, castor and cassava in Euphorbiaceae are also very important cash crops. Both rubber tree and castor have high oil content, while cassava is an important source of starch in the world. Unfortunately, each inflorescence in these species produces only a small number of female flowers, too [22,23,24]. As mentioned earlier, *LBD* genes have very important roles in pollen development, inflorescence development, and they affect a variety of organ development. Thus, it is important to identify and analyze the *LBD* gene family of Euphorbiaceae.

With the continuous development of sequencing technology, the draft genome of several species of Euphorbiaceae have been released [25,26,27,28]. It is now easy for us to further understand the physiological and biochemical processes of Euphorbiaceae plants through gene family analysis. In this study, we first identified 33, 58, 54 and 30 *LBD* members of *J. curcas*, rubber tree, cassava and castor, respectively. Next, we conducted a comprehensive analysis of basic physicochemical information, phylogenetic analysis, gene structure and protein motif analysis, *cis*-element prediction, miRNA target site prediction and gene duplication event analysis on them. Finally, we focused on *J. curcas* for further expression analysis, and downstream target gene prediction and annotation to provide theoretical reference for the study of the role of *LBD* genes in the sexual differentiation of *J. curcas*.

## 2. Results

### 2.1. Genome-Wide Identification of LBD Genes in J. curcas, Rubber Tree, Cassava and Castor

Based on hmmsearch and BLASTP, 33, 58, 54 and 30 *LBD* genes were identified in *J. curcas*, rubber tree, cassava and castor, respectively, and named as *JcLBD1*-*JcLBD33*, *HbLBD1*-*HbLBD58*, *MeLBD1*-*MeLBD54* and *RcLBD1*-*RcLBD30* according to their positions on chromosomes or scaffolds.

Basic information about Euphorbiaceae LBD proteins were calculated (listed in Appendix A). The proteins encoded by the 33 *LBD* genes of *J. curcas* with amino acid lengths ranged from 119 (JcLBD2) to 332 (JcLBD31); molecular weights ranged from 13,481.67 (JcLBD2) to 37,487.12 (JcLBD31), except the JcLBD3, which had a length of 1077 amino acids and a molecular weight of 117,788.7 Da; the predicted isoelectric points of the JcLBD proteins ranged from 5.1 (JcLBD17) to 9.28 (JcLBD9); the fatty amino acid index ranged from 61.38 to 95.21, indicating a small difference in their thermal stability; and the minimum instability index was 43.75 (JcLBD10), indicating that they were all unstable in vitro; the hydrophilicity scores of all proteins were smaller than 0, indicating that they were all hydrophilic. The lengths of the proteins encoded by the *LBD* genes of rubber tree, cassava and castor ranged from 116–333, 116–302, 117–310 amino acids, respectively; the molecular weights ranged from 13,029.18–37,240.79, 13,138.24–33,889.14, 13,033.05–34,490.75; the isoelectric points ranged from 5.02–9.32, 4.67–9.02, 4.53–9.45; fatty amino acid index (A.I.) indices ranged from 51.94–90.86, 60.76–88.06, 57.78–91.89, respectively; all proteins had instability indices greater than 40, except for MeLBD35 and RcLBD13, which had instability indices of 36.04 and 38.72, and were unstable in vitro; the hydrophilicity scores of all proteins were less than 0, indicating that they were all hydrophilic. These results showed that LBD proteins are relatively conserved in these four species of Euphorbiaceae in terms of physical properties.

### 2.2. Phylogenetic Analysis

To better understand the evolutionary trajectory and function of the Euphorbiaceae *LBD* gene family, we constructed a phylogenetic tree using 175 identified LBD proteins in Euphorbiaceae and 43 known LBD proteins in *A. thaliana* (Figure 1). The Euphorbiaceae *LBD* gene family can be divided into two subfamilies according to the presence or lack of the motif LX6LX3LX6L. Most of the *LBD* genes belong to Class I, which contained 183 (83.94%) *LBD* genes, while Class II contained only 35 (16.06%) *LBD* genes. Further analysis found that *LBD* genes of Euphorbiaceae could also be divided into seven subclasses, Class Ia-Ie, Class IIa and Class IIb. Among these subclasses, Class Ia contained the largest number of *LBD* gene family with 45 members, and Class Ic had the least with only 12 genes. Each subclass contained *LBD* genes of all species, suggesting that they may share a common ancestor. Most of the LBD members of Euphorbiaceae were clustered together. LBD members of *A. thaliana* were clustered together as well, such as AtLBD10, AtLBD26, AtLBD28, AtLBD32 and AtLBD35 of Class Ia. This result suggests that *LBD* genes in Euphorbiaceae and *A. thaliana* diverged during evolution.

### 2.3. Gene Structure and Protein Motif Analysis

To further explore the possible evolutionary relationship and function of *LBD*s in Euphorbiaceae, we performed gene structures and protein motifs analyses of them (Figure 2 and Appendix A). The number of introns in the *LBD* genes of Euphorbiaceae ranged from 0 to 14. Of those, 43 (24.57%) *LBD* genes of Euphorbiaceae did not contain a intron, 113 genes (64.57%) contained 1 intron, 18 genes (10.29%) contained 2 introns, and only *JcLBD3* contained 14 introns. *LBD* genes close in the phylogenetic tree tended to exhibit similar exon-intron structures: most Class Ia *LBD* genes did not contain an intron; most Class Ib, Id, IIa and IIb LBD genes contained only one intron; all Class Ic genes contained two introns; about half of Class Ie genes contained no intron, and another half contained one intron.

An analysis of protein motif results showed that all Euphorbiaceae LBD proteins contained CX2CX6CX3C zinc finger motif (motif 1 in Ia-Ic, Ie and IIa; motif 2 for IIb; motif 1 and motif 3 for Id). The Euphorbiaceae LBD proteins in the same clade contained similar motifs, indicating that the classification of the Euphorbiaceae *LBD* gene family in this study is reliable. The LBD proteins of Euphorbiaceae and *A. thaliana* in the same clade usually contain different motifs, indicating that the *LBD* genes of Euphorbiaceae and *A. thaliana* may be functionally differentiated.

### 2.4. Cis-Element Prediction

The analysis of *cis*-elements is the key to understanding gene regulation patterns, and provides important information for further understanding the function of genes. We have submitted the 1500 bp upstream sequences of Euphorbiaceae *LBD* genes to the plantCARE database for *cis*-element search. Of these *cis*-elements, 23 representative elements were extracted for display (Figure 3 and Appendix A). These 23 *cis*-elements were related to hormone response, stress response and plant development. The type and number of *cis*-acting elements in Euphorbiaceae and *Arabidopsis LBD* genes did not differ significantly. Among these 23 components, the ARE element associated with hypoxic stress was the most abundant. Almost all *LBD* promoters contain ARE element: 34, 23, 45, 34 and 22 in *A. thaliana*, *J. curcas*, rubber tree, cassava and castor, respectively. Ethylene (ERE), abscisic acid (ABRE), drought (MBS), stress (STRE), and light- (G-box) related response elements also appeared in the promoter regions of several genes. In addition, the *LBD* promoters contained the hormone-related response elements of jasmonic acid (CGTCA-motif), salicylic acid (TCA-element), auxin (AuxRR-core and TGA-element), gibberellin (P-box, GARE-motif and TATC-box); stress response elements of wound (WUN-motif), low temperature (LTR), defense and stress (TC-rich repeats), anoxic inducibility (GC-motif), and light (GT1-motif); and developmental regulatory elements: circadian, HD Zip 1, CAT-box and GCN4_motif. The types and numbers of *cis*-elements of different *LBD* genes in the same species are quite different, even for genes located in the same clade. These results suggested that *LBD* genes may respond to different signaling pathways due to different *cis*-acting elements in its promoter region.

### 2.5. Prediction of miRNA Target Sites

The prediction of miRNA target sites of genes would provide further understanding of the regulation model of the genes. We submitted the CDS sequences of *LBD* genes of each species together with mature miRNAs to psRNATarget to predict miRNA target sites. It was predicted that 30, 8, 24, 19 and 6 *LBD* genes are regulated by miRNA in *A. thaliana*, *J. curcas*, rubber tree, cassava, and castor, respectively. And they are regulated by several different miRNA families (Figure 4 and Appendix A). All species have members of the *LBD* gene family regulated by miRNA172, in which miRNA172 mainly regulated members of the Class Ia, Id, and Ie *LBD* genes in five species. In addition, miRNA156, miRNA159, and miRNA164 can also act on multiple *LBD* genes in Euphorbiaceae. Genes in the same subfamily may be regulated by different miRNAs. These results suggested that the interaction between *LBD* genes and miRNAs in Euphorbiaceae is not conservative, and the homologous genes may have evolved in different regulatory patterns.

### 2.6. Gene Duplication and Selective Pressure Analysis

To further investigate the evolution of *LBD* genes in the Euphorbiaceae, we performed gene duplication analysis and selection pressure analysis. We first used MCScanX software to analyze duplicated *LBD* genes of each species in Euphorbiaceae. As listed in Appendix A, all *LBD* genes in *J. curcas*, rubber tree, cassava and castor were duplication genes. Among them, 0, 2, 0, 2 of the LBD genes in *J. curcas*, rubber tree, cassava, and castor were proximal duplication genes; 5, 6, 4, 1 were tandem duplication genes; 15, 12, 42, 7 were segmental duplication genes respectively. The number of segmental duplication genes was significantly higher than that of proximal duplication genes. This suggests that segmental duplication may be important for the *LBD* gene family.

Next, we analyzed the selection pressure by calculating the Ka and Ks of gene duplication pairs (Figure 5 and Appendix A). We identified 64 duplication gene pairs in Euphorbiaceae: 3, 3, 4 and 0 tandem duplication genes and 9, 6, 35 and 4 segmental duplication gene pairs in *J. curcas*, rubber tree, cassava, and castor, respectively. The Ka/Ks of all duplicated gene pairs ranged from 0.046269 to 0.389515, which was much less than 1, suggesting that they were subjected to purifying selection during the evolutionary process. From Ks value, we can see that the *LBD* genes of Euphorbiaceae experience two large-scale duplication events. The most recent duplication event mainly involved the duplication of *HbLBD*s and *MeLBD*s. Previous studies discovered that rubber tree and cassava experienced another whole genome duplication event compared to other Euphorbiaceae species. And our results are consistent with this conclusion [29].

### 2.7. Collinearity Analysis

To infer the evolutionary relationship of *LBD* genes among different species, we performed a collinearity analysis on the Euphorbiaceae genomes. Since the rubber tree and castor genomes were not assembled to the chromosome level, only *J. curcas*, cassava and *A. thaliana* genomes were selected for analysis. As shown in Figure 6, there were many collinear blocks between them. *J. curcas* and *A. thaliana* have 15 collinear blocks containing *LBD* genes, and they contained a total of 17 *LBD* gene pairs; *J. curcas* and cassava have 25 collinear blocks containing a total of 40 *LBD* gene pairs. In addition, most of *J. curcas* had more than two orthologs in cassava, suggesting that cassava has undergone an additional whole genome duplication event during its evolution.

### 2.8. Gene Expression Analysis of JcLBD Genes

To further investigate the potential functions of each *LBD* gene in *J. curcas*, we analyzed the expression of *JcLBD*s in different tissues and stress treatments. Figure 7 and Appendix A show the tissue expression profile of *JcLBD* genes. We found that the genes of the same subclass usually have similar expression profiles. Class Ia *LBD* genes were mainly expressed in flowers; class Ib and Ic genes were mainly expressed in roots; class Id genes are mainly expressed in roots and fruits; class Ie genes were mainly expressed in flowers, especially male flowers; and in class II, there were no significant differences in gene expression among the organs, except *JcLBD29*, which was highly expressed in fruits. *JcLBD6*, *JcLBD7*, and *JcLBD8* are tandem duplication genes, and they were extremely close on the phylogenetic tree. *JcLBD6* and *JcLBD7* have samilar expression profiles and both expressed highly in male flowers; *JcLBD11* clustered on the same phylogentic branch with *JcLBD6* and *JcLBD7*, and it was also expressed in male flowers; however, *JcLBD8* was expressed mainly in fruit, and it also expressed in male flower, suggesting that genes on this evolutionary branch may be extremely important for male flower development. *JcLBD1* and *JcLBD21* are two proteins produced by segmental duplications, both of which have high expression in male flowers, suggesting that they are likely to play a very important role for male flower development. *JcLBD18*, *JcLBD27*, and *JcLBD32* are close on the phylogenetic tree, and *JcLBD27* and *JcLBD32* are segmental duplication genes that are significantly expressed in female flowers, indicating that they may be associated with female flower development. Moreover, we found these genes have similar expression profiles and protein motifs, but their *cis*-acting elements and miRNA interactions are relatively different. These results indicate that these genes play extremely important roles in the growth and development of *J. curcas*, and they have evolved different regulatory patterns despite their conserved functions.

The Figure 8A,B are the expression profiles of the *JcLBD* genes after drought and salt stress treatments, respectively. The heatmap reveals that neither salt stress nor drought treatments obviously changed the expression of the majority of *JcLBD*s in the leaves. But in root, the situation is very different; the response of *LBD*s to stress gets more complex. Since there are only two replicates per sample, our expression analysis below was based on fold change.

As shown in Figure 8A and Appendix A, the expression of *JcLBD14* increased in leaves at day 1 after drought treatment (2 fold change); *JcLBD7* and *JcLBD32* were highly expressed in roots at day 4 after drought treatment (70 and 7 fold change in *JcLBD7* and *JcLBD32*, respectively); *JcLBD19* showed increased expression in both leaves and roots at day seven after drought treatment (34 fold change in leaves and 50 fold change in roots); and drought treatment results in altered timing of *JcLBD17* and *JcLBD25* expression in roots (both highly expressed in C4dR and D1dR).

As shown in Figure 8B and Appendix A, *JcLBD25* had low expression in control leaves (2.168, 15.5785, and 11.4697 in C2hL, C2dL, and C7dL, respectively), but after salt stress treatment, its expression in leaves was 0. *JcLBD19* and *JcLBD29* also showed decreased expression in roots after salt stress treatment (20 and nine-fold change after 2 h in *JcLBD19* and *JcLBD29*, respectively); most salt response genes in the roots showed increased expression after salt treatment, such as *JcLBD8*, *JcLBD15*, and *JcLBD17* (five- and three-fold change after two days and seven days in *JcLBD8*, three- and four-fold change after 2 h and two days in *JcLBD15*, and a five-fold change after 2 h in *JcLBD17*).

These results indicated that the *JcLBD*s are mainly expressed in roots, flowers and fruits, and that *JcLBD*s barely respond to drought and salt stress in leaves, while some *JcLBD*s expressions altered in roots.

### 2.9. Prediction and Annotation of Target Genes

Finally, we screened the downstream target genes that may be regulated by JcLBD transcription factors, and annotated the function of these target genes with GO enrichment and expression analysis in order to further investigate the potential roles of *LBD*s in *J. curcas*. Table 1 showed the best possible binding sites for three known JcLBDs and their corresponding motif information. Using the binding motif scanning method, we identified 148, 18, and 69 target genes that may be regulated by JcLBD27, JcLBD24 and JcLBD22, respectively (Appendix A).

As shown in Figure 9, the GO annotation results showed that JcLBD27 target genes’ biological processes were mainly meristem initiation (GO:0010014), meristem structural organization (GO:0009933), secondary shoot formation(GO:0010223), shoot axis formation (GO:0010346) and morphogenesis of a branching structure (GO:0001763), anatomical structure arrangement (GO:0048532); and the biological processes of JcLBD24 genes were mainly the regulation of DNA-binding transcription factor activity (GO:0051090), protein-DNA complex subunit organization (GO:0071824); and the JcLBD22 were mainly secondary shoot formation (GO:0010223), shoot axis formation (GO:0010346), morphogenesis of a branching structure (GO:0001763), and meristem initiation (GO:0010014). The tissue expression profiles of the JcLBD target genes were displayed in Appendix A, and we can see that these genes were highly expressed primarily in flowers, leaves and roots. These results indicate that *JcLBD*s may play a very important regulatory role in plant growth and development.

## 3. Discussion

The *LBD* gene family is a plant-specific transcription factor family which plays an important role in the growth and development of various lateral organs and stress responses [4,16]. *LBD* genes are widely distributed in the plant kingdom, from green algae to angiosperms, and have been identified and studied in a variety of plants including *A. thaliana*, rice, moso bamboo, wheat, ginkgo, potato, fassion fruit, and so on [3,30,31,32,33,34,35]. However, the *LBD* genes have not been studied in Euphorbiaceae yet. The high-quality genomes of *J. curcas*, cassava, castor and rubber tree make it possible for us to explore the function of *LBD* genes in Euphorbiaceae. In this study, we identified *J. curcas*, rubber tree, cassava and castor containing 33, 58, 54 and 30 LBD family members, respectively. Previous studies identified 28 *LBD* genes in *Physcomitrium patens*, 71 in *Picea abies*, 44 in maize, 43 in *A. thaliana*, 42 in grapes, and 55 in *Eucalyptus grandis* [11,36]. This suggested that the *LBD* gene family retained largely function in the genetic evolution of the angiosperms species. The *LBD* gene family of Euphorbiaceae can be divided into seven subclasses (Ia-Ie and IIa-IIb). Although the phylogenetic tree classification of this family of Euphorbiaceae is similar to that of *A. thaliana*, members of the family of Euphorbiaceae appear clustered to one branch. The results of protein motif analysis showed that proteins close on the phylogenetic tree have similar structures, but LBD proteins of Euphorbiaceae have evolved new motifs compared to *A. thaliana*. Gene structure analysis showed that the intron numbers and length of *LBD* genes of Euphorbiaceae is more variable than those of *A. thaliana*. For example, *JcLBD3* of the Ib subfamily contains 14 introns, and the intron length of *HbLBD3*-*HbLBD26* of Ie subfamily is significant longer than others. In addition, there are 64 pairs of LBD duplication genes in Euphorbiaceae, of which 10 pairs are tandem duplication genes and 54 pairs are fragment duplication genes. The collinearity blocks between *J. curcas* and cassava *LBD* genes were significantly more than those between *J. curcas* and *A. thaliana*. These results suggested that the *LBD* genes function may be conserved in Euphorbiaceae.

Promoter *cis*-acting element prediction, expression pattern analysis, and target gene prediction and annotation can better explain the possible functions of *JcLBD*s. The results of *cis*-element analysis showed that the elements contained in Euphorbiaceae *LBD* genes are mostly related to ethylene, drought and hypoxia responses. *LBD* genes located in the same clade of *J. curcas* usually contained different *cis*-acting element types and quantities. The miRNA prediction results showed that miRNA156, miRNA159, miRNA164, and miRNA172 can target multiple *LBD* genes in Euphorbiaceae, and they all can regulate plant development [37,38,39]. Except these four miRNAs, the miRNA families interacting with *LBD* genes in different species of Euphorbiaceae were quite different, and *LBD* genes in the same subclass may be regulated by different miRNA family members. Target gene prediction and annotation results showed that the downstream target genes of *JcLBD*s play roles in various biological processes. These results suggest that LBD family members of *J. curcas* may play regulatory roles in various signaling pathways through complex synergistic effects, thereby participating in various physiological processes.

Most of the Euphorbiaceae species are dioecious, and the ratio of female to male flowers is very low. Therefore, it is very important to study the regulation of flower development in Euphorbiaceae. According to the tissue expression pattern in Figure 7, we found that multiple members of the Ia and Ie subfamily are specifically expressed in male or female flowers. Previous studies have suggested that the Ia subfamily has an important function in the formation of aerial organs, while the Ie subfamily function has not yet been summarized [16]. *JcLBD6*, *JcLBD7*, and *JcLBD8* in Class Ia are close in the phylogenetic tree, and they are tandem duplication genes. *JcLBD6* and *JcLBD7* are clustered in one clade, and *JcLBD11* is on this clade, as well. *JcLBD6* and *JcLBD11* are maily expressed in male flower, and *JcLBD7* is expressed in male flower and root. According to the phylogenetic analysis, *JcLBD6*, *JcLBD7*, *JcLBD8* and *JcLBD11* are homologous genes of *AtLBD36*/*AS1*. *AS1* can regulate flower development [40,41]. This indicates that the function of *AtLBD36* may be separated in *J. curcas*, and *JcLBD6*, *JcLBD7* and *JcLBD11* may be extremely important for the male flower development of *J. curcas*. *JcLBD1* and *JcLBD21* are segmental duplication genes in Class Ie, and they are both highly expressed in male flowers. Their homologous protein *AtLBD27* has been confirmed to play an extremely important role in pollen development. The *lbd27* mutant causes pollen abortion (the abortion rate was high as 70%), and all pollen aborted in *lbd10 lbd27* double mutant [9]. This clade contains another *A. thaliana* gene, *AtLBD22*, which has also been shown to play a role in pollen development [42]. In addition, the types and numbers of *cis*-acting elements of *JcLBD1* and *JcLBD21* are quite different, and *JcLBD1* is a target site of miRNA172, so it is speculated that *JcLBD1* and *JcLBD21* are also involved in the development of pollen. It is speculated that the Euphorbiaceae LBD proteins of this clade may play a regulatory function on pollen development through different pathways. *JcLBD18*, *JcLBD27* and *JcLBD32* are close in the phylogenetic tree, and *JcLBD27* and *JcLBD32* are segmental duplication genes, which are significantly expressed in female flowers. The phylogenetic tree shows that *JcLBD27* and *JcLBD32* are homologous of *AtLOB*, which has been confirmed to play an important role in the development of *A. thaliana* lateral organs, and several important homologous proteins identified in other species. *OsRA2*, rasoma2, and *HvRA2* are homologous of *AtLOB,* which is involved in the regulation of floral development, so it is speculated that *JcLBD27* and *JcLBD32* may also regulate flower development [3,6,7,8]. In addition, *JcLBD9* and *JcLBD28* of the Class Ie are divided into two major branches on the phylogenetic tree, but they are all significantly expressed in flower. These results suggest that Class Ia and Class Ie proteins in *J. curcas* are likely to play an important role in the regulation of flower development in plants, and appear to be more delicately regulated than *AtLBD*s. The functional differentiation of *JcLBD*s may occur due to transcriptional regulation and post-transcriptional modifications. For example, *JcLBD6*, *JcLBD7*, *JcLBD8* and *JcLBD11* promoters contain different *cis*-acting elements; *JcLBD5*; *JcLBD16* and *JcLBD23* are regulated by miRNAs in different ways: *JcLBD5* and *JcLBD16* are regulated by miR159 and miRN1624, respectively, while *JcLBD23* is not regulated by miRNAs.

## 4. Materials and Methods

### 4.1. Collection of Sequencing Data

Genome data and protein data of *Jatropha curcas* come from the Giga database (http://gigadb.org/dataset/view/id/100689, accessed on 8 December 2021); cassava data was downloaded from the Phytozome V13 database (Manihot esculenta v6.1; https://www.ncbi.nlm.nih.gov/, accessed on 29 August 2021); the gene family information of *A. thaliana* was downloaded from the TAIR database (https://www.arabidopsis.org/, accessed on 2 June 2021); and the genome and protein data of *A. thaliana* (TAIR10.1), rubber tree (ASM165405v1), and castor (JCVI_RCG_1.1) were downloaded from the NCBI database (https://www.ncbi.nlm.nih.gov/, accessed on 29 August 2021).

### 4.2. Identification of LBD Genes

In order to identify the *LBD* gene family members of *J. curcas*, rubber tree, cassava and castor, a hidden Markov model of the LOB domain (PF03195) was obtained from the Pfam database (http://pfam.xfam.org/, accessed on 28 March 2022) and used as the seed model for an HMMER3 search of the local Euphorbiaceae protein database [43]. In addition, 43 published *A. thaliana* protein sequences containing the LOB domain were used as the original alignment sequence of BLASTP [44]. The sequences identified by these two methods were submitted to NCBI CD-Search (http://www.ncbi.nlm.nih.gov/Structure/cdd/wrpsb.cgi, accessed on 29 March 2022) to confirm the conserved domain [45]. Genes with incomplete LOB domain and redundant genes were removed to produce the confirmed LBD genes.

LBD genes of Euphorbiaceae were renamed according to their positions on the chromosomes, and then the ExPASY tool (https://web.expasy.org/protparam/, accessed on 18 April 2022) was used to predict protein physicochemical parameters such as protein size (aa), molecular weight (MW), isoelectric point (PI), stability, fatty amino acid index (A.I.), and the grand average of hydropathicity (GRAVY) [46].

### 4.3. Phylogenetic Analyses

Multiple sequence alignment of all LBD protein sequences of Euphorbiaceae and *A. thaliana* were by Clustal W. And based on the alignment results, maximum likelihood (ML) trees were constructed using MEGA X software with the JTT model, and the Bootstrap was set to 1000 times [47]. The phylogenetic tree was visualized and modified using the online website Evolview (https://www.evolgenius.info/evolview/#/treeview, accessed on 14 June 2022) [48].

### 4.4. Motif and Gene Structure Analysis

The MEME suite was used for conservative motif prediction with the following parameters: maximum width 50, minimum width 6, and the number of motifs set to 20 [49]. The results were then further modified with the TBtools software [50]. To analyze gene structure, we first extract exon and intron positions from the gene annotation file and then submit them to the online Gene Structure Display Server (GSDS: http://gsds.gao-lab.org/, accessed on 13 April 2022) for visualization [51].

### 4.5. Cis-Acting Element Analysis

The 1500 bp sequence upstream the *LBD* genes were extracted as promoter sequences based on the gene annotation and chromosome sequence, and the plantCARE database (http://bioinformatics.psb.ugent.be/webtools/plantcare/html/, accessed on 4 May 2022) was used to predict the *cis*-element [52]. The elements related to hormone response, stress response and plant developmrnt were then extracted. R package pheatmap was used for further visualization [53].

### 4.6. miRNA Target Gene Analysis

Mature miRNA sequences of the Euphorbiaceae species were downloaded from the plant microRNA Encyclopedia Database (PmiREN; https://www.pmiren.com/, accessed on 18 April 2022), and CDs sequences of the *LBD* gene were inputted into the psRNATarget online tool (https://www.zhaolab.org/psRNATarget/analysis, accessed on 28 April 2022) to predict the target sites with the default parameters [54,55]. The interacting miRNAs and target genes with expectation values greater than 4.5 were extracted, and Cytoscape software was used to construct the interaction network between these miRNAs and the target genes [56].

### 4.7. Collinearity and Selective Pressure Analysis

Collinearity of *LBD* genes in Euphorbiaceae was analyzed by JCVI [57]. Since the genome of the rubber tree and castor were only assembled to the scaffold level, we only analyzed the collinearity between species for *J. curcas*, cassava, and *A. thaliana*.

Gene duplication includes the tandem duplication and segmental duplication. First, BLASTP was used to all-against-all BLAST, and the results were used to identify gene duplication events using MCScanX v1.1. Finally, KaKs_ Calculator2.0 was used to calculate the synonymous substitution rate (Ks), the nonsynonymous substitution rate (Ka), and the Ka/Ks ratio between homologous gene pairs [58].

### 4.8. Tissue Expression Analysis and Stress Response Analysis of JcLBDs

To further explore the potential function of *JcLBD*s in *J. curcas*, we examined the expression of *JcLBD*s through public transcriptome data. Raw data of six different tissues including fruit, male flower, female flower, leaf, root, and stem were obtained for tissue expression analysis (listed in Appendix A; Accession number: PRJNA399175). Raw expression data of *J. curcas* treated by drought and salt were used to analysis, too (listed in Appendix A; Accession number: PRJNA257901 and PRJNA244896). Hisat2 and StringTie were used for comparison and quantitative analysis, respectively [59,60]. GCEN softwere were then used for normalization by quantile normalization algorithm [61]. For samples with duplicates, we calculated the avarage of TPM values. A heatmap of different tissues and stress treatment of the *JcLBD*s was drawn by R package pheatmap using log10(TPM) values [53].

### 4.9. Identification and Annotation of Downstream Genes

To obtain potential downstream regulatory genes for the LBD protein, we used bedtools to extract 1500 bp of the *J. curcas* promoter sequence as potential target sites for binding.

The DNA binding sites of three members of the *JcLBD* genes are already known, and the base sequences of these three binding sites were downloaded from the database (http://planttfdb.gao-lab.org/index.php, accessed on 2 June 2022) [62]. Then the Motif FIMO program (5.3.0) were used to detect the binding site with the P  <  1 × 10^−7^ [49]. Genes containing binding sites were then considered to be downstream target genes of LBD.

The eggNOG-mapper was used for GO annotation of all target genes of *JcLBD*s [63]. Then GO enrichment of downstream target genes for each of the three known binding site genes was performed using the R package clusterProfiler [64]. Tissue-specific expression analysis of these target genes was performed separately using the R package pheatmap (listed in Appendix A; Accession number: PRJNA399175) [53].

## 5. Conclusions

In this study, the members of the Euphorbiaceae *LBD* gene family were identified and analyzed for the first time by means of bioinformatics. *Jatropha curcas*, rubber tree, cassava and castor contained 33, 58, 54 and 30 LBD members, respectively. Phylogenetic analysis found that Euphorbiaceae *LBD* genes could be divided into seven subgroups. The properties of LBD proteins in Euphorbiaceae species are similar, and similar proteins on the Euphorbiaceae phylogenetic tree have similar gene structures and protein motifs, indicating that the *LBD* genes of Euphorbiaceae are conserved in the evolutionary process. The number of LBD members in rubber tree and cassava is twice that of *J. curcas* and castor, and most of the replicative gene pairs are generated by fragment duplication. Further analysis of *cis*-acting elements, miRNA target sites, expression profiles, protein interactions, and target gene prediction and annotation analysis of *JcLBD*s showed that Class Ia and Ie genes have important regulatory effects on plant flower development. This study provides a reference for further exploration of the mechanism of sexual differentiation in *Jatropha curcas*.

## Figures and Tables

**Figure 1 plants-11-02397-f001:**
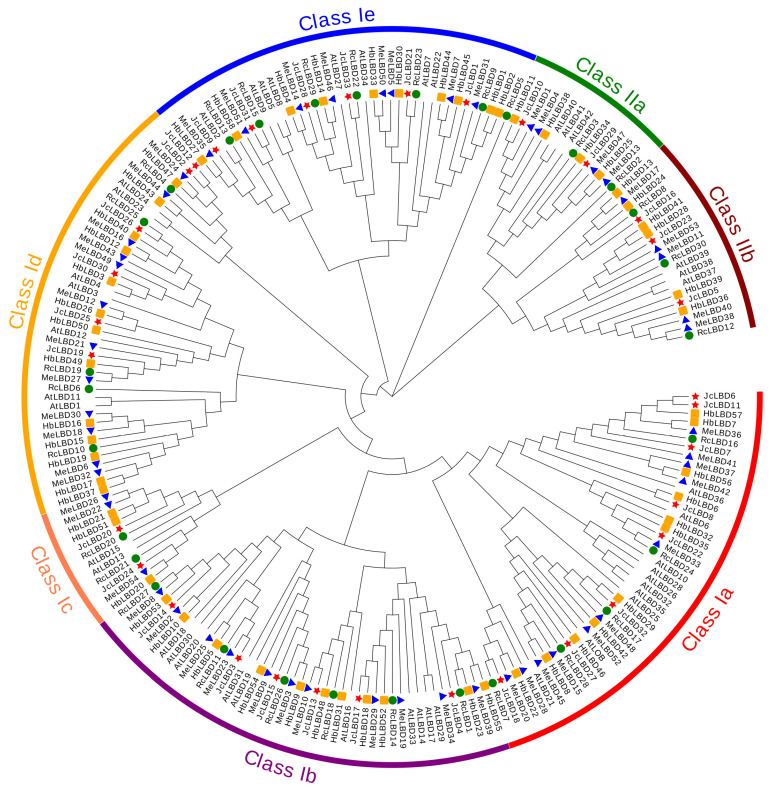
The phylogenetic tree of *LBD* genes of *J. curcas*, rubber tree, cassava, castor and *A. thaliana* was constructed by MEGA 10 using the maximum likelihood (ML) model. Red stars, orange rectangles, blue triangles, and green circles indicate *J. curcas*, rubber tree, cassava, and castor sequences, respectively.

**Figure 2 plants-11-02397-f002:**
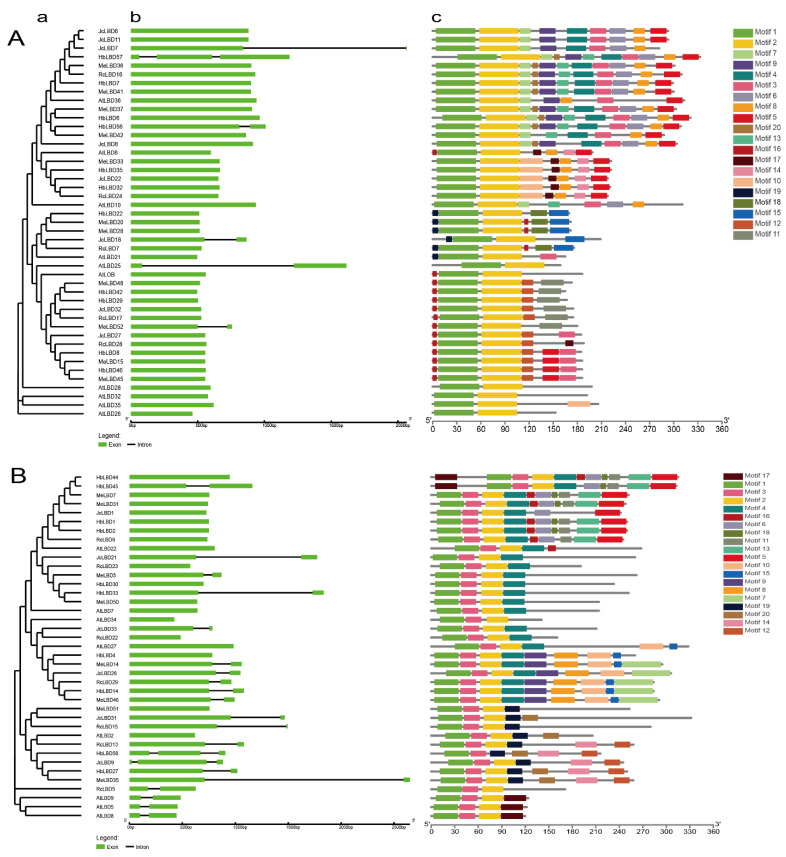
The phylogenetic relationship, gene structure, and conserved protein motifs of Euphorbiaceae *LBD* genes. (**A**,**B**) are phylogenetic tree, gene structure and conserved motif of Class Ia and Ie of Euphorbiaceae, respectively. (a) A maximum likelihood tree was constructed by MEGE 10 with 1000 bootstrap replicates; (b) Exon-introns structure of *LBD*s performed by GSDS2; (c) The conserved motifs of Euphorbiaceae species LBD proteins analysed by MEME suit.

**Figure 3 plants-11-02397-f003:**
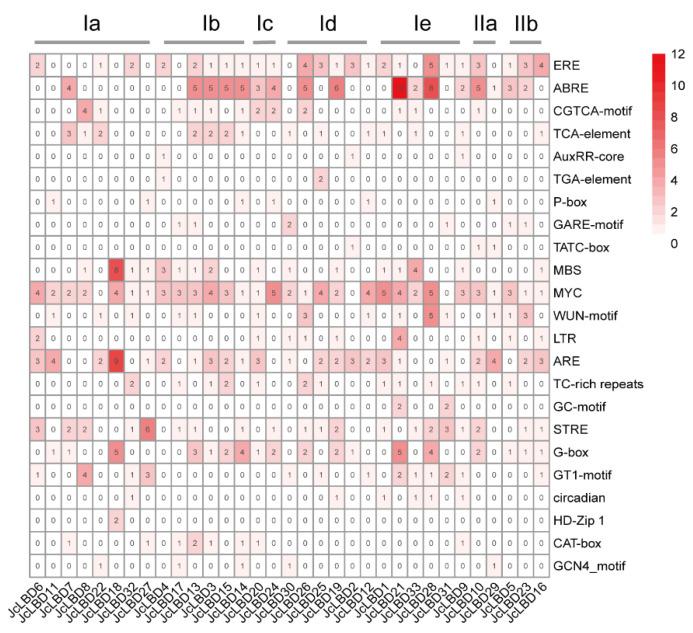
*Cis*-acting element matrix identified in the 1500-bp upstream promoters of each *JcLBD*s.

**Figure 4 plants-11-02397-f004:**
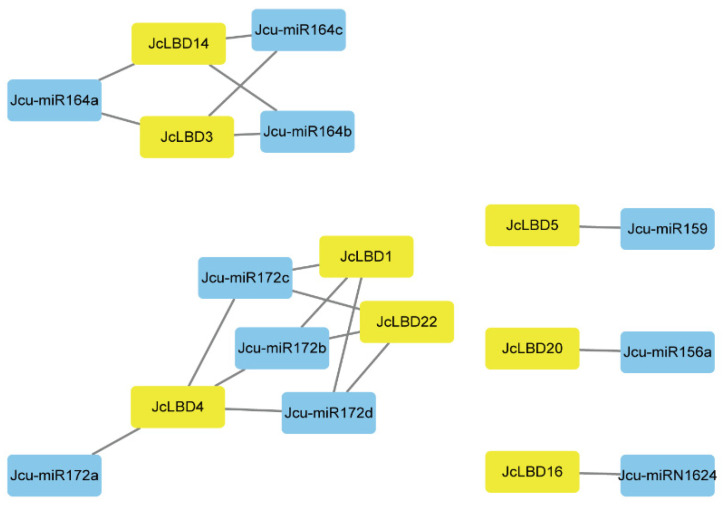
Netwrok between the Jcu-miRNAs and their targeted *JcLBD* genes. The blue and yellow rectangles indicate miRNAs and target genes, respectively.

**Figure 5 plants-11-02397-f005:**
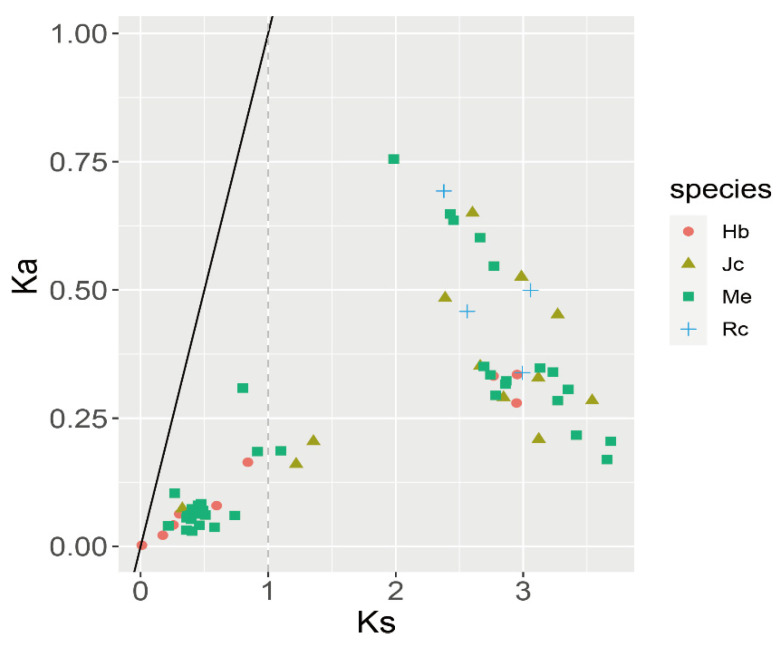
Selective pressure of Euphorbiaceae species *LBD* duplication genes. Red circles, yellow triangles, green rectangles, and blue crosses represent *LBD* duplication gene pairs of rubber tree, *J. curcas*, cassava, and castor, respectively.

**Figure 6 plants-11-02397-f006:**
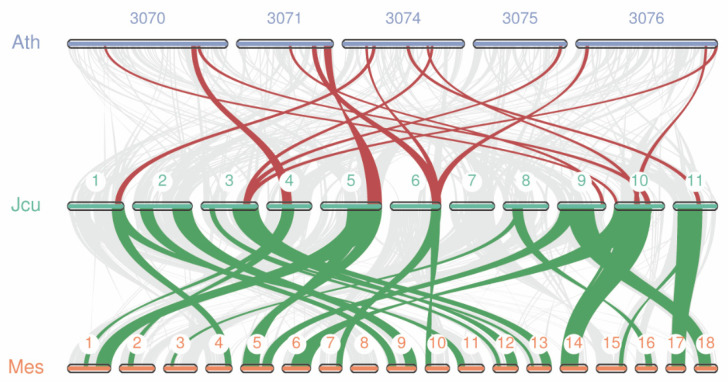
*LBD* genes synteny analysis among the three species *J. curcas*, cassava, and *A. thalina*. The grey lines on the background indicate the collinear blocks within *J. curcas* and other species genomes, and the hilighted lines are the collinear blocks containg the *LBD* genes. Ath, Jcu, and Mes represents *A. thalina*, *J. curcas*, and cassava, respectively. 3070, 3071, 3074, 3075, and 3076 respresnt chromsomes NC_003070.9, NC_003071.7, NC_003074.8, NC_003075.7, and NC_003076.8, respectively. 1–11 in green represent chromosomes chr1–chr11, respectively. 1–18 in orange represent chromosomes Chromosome01–Chromosome18, respectively.

**Figure 7 plants-11-02397-f007:**
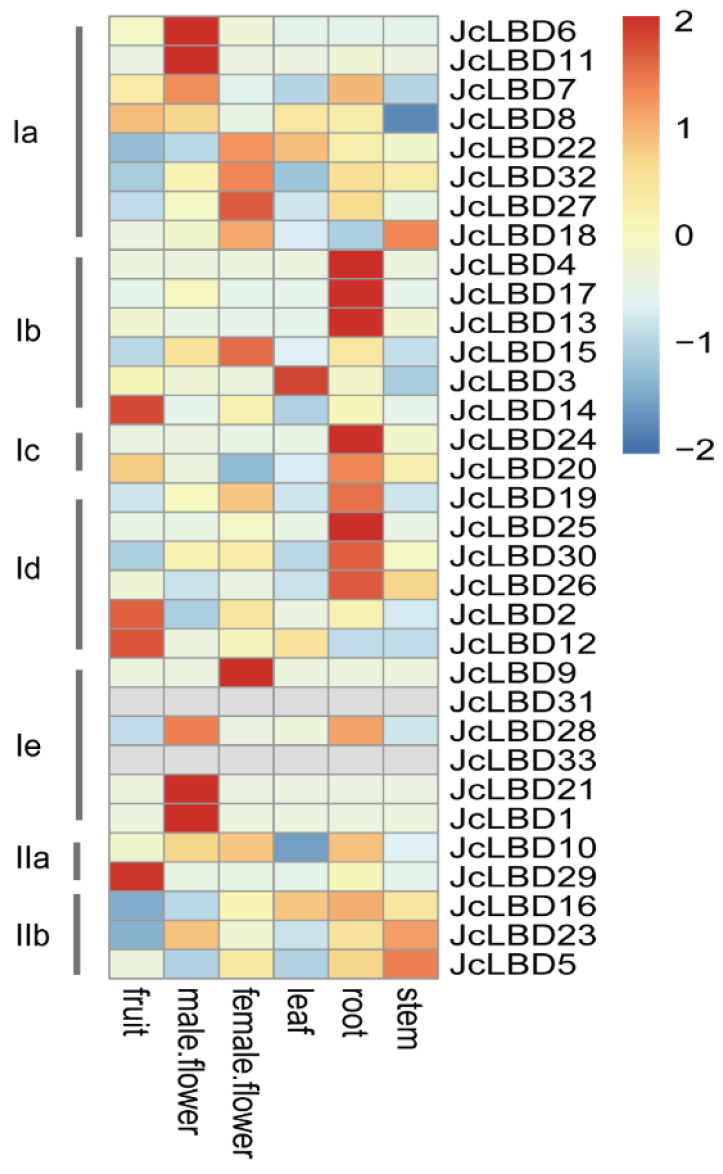
Tissue expression patterns of *JcLBD*s. Heatmap of expression levels of 33 *JcLBD* genes in different tissues. fruit: immature fruit; male flower: fully open; female flower: fully open; leaf: fully mature; root: fully expanded; stem: fully mature. Heatmap was drawn by pheatmap using log10(TPM) values.

**Figure 8 plants-11-02397-f008:**
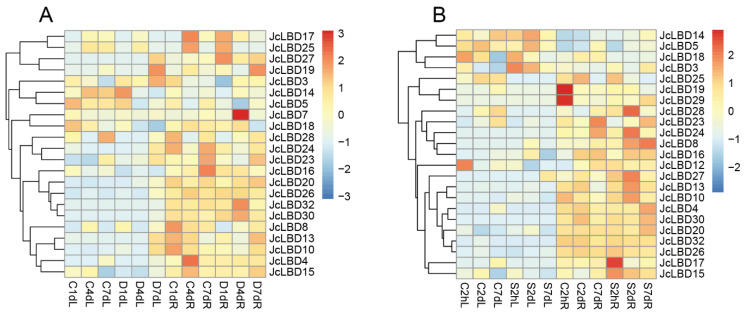
Stress response analysis of *LBD*s in *Jatropha curcas*. (**A**) Expression of *JcLBD*s in drought treatment. C and D represent control and drought treatments, respectively, and 1d, 4d and 7d correspond to one, four, and seven days post-treatment sampling, respectively, and L and R represent leaf and root tissues, respectively. (**B**) Expression of *JcLBD*s under salt stress. C and S represent control and salt stress treatments, respectively, 2h, 2d and 7d correspond to 2 h, two days and seven days post-treatment sampling, respectively, and L and R represent leaf and root tissues, respectively. Heatmap was drawn by pheatmap using log10(TPM) values.

**Figure 9 plants-11-02397-f009:**
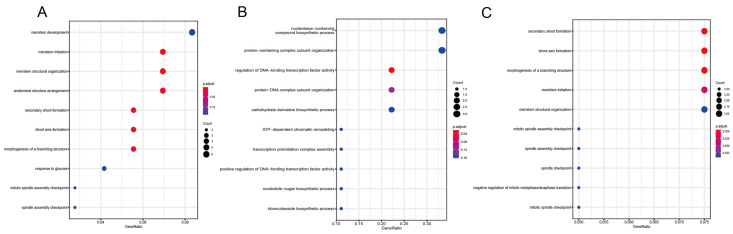
GO enrichment of JcLBD target genes. (**A**–**C**) are the downstream target genes of JcLBD27, JcLBD24 and JcLBD22, respectively.

**Table 1 plants-11-02397-t001:** Best possible binding sites for JcLBD*s*.

Transcription Factor	Motif	Best Possible Match
JcLBD27	Jcr4S00009.60	TCCGCCGCCGCCTCCGCCGCC
JcLBD24	Jcr4S00803.60	CGGCGGAAATTGCGGCG
JcLBD22	Jcr4S13769.10	TCTCCGCCGCCTTCTCCGCCG

## Data Availability

Not applicable.

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
