# Peer review of "Genome-Wide Identification and Analysis of Lbd Transcription Factor Genes in Jatropha curcas and Related Species"

_plants, 2022, doi:10.3390/plants11182397_

Round 1
Reviewer 1 Report
This manuscript provides an in-silico analysis of the structure and putative functions of LBD transcription factor genes in Euphorbiaceae. Although this study has been carried out in other species, new data on the molecular structure of this type of transcription factors are provided, as well as their possible involvement in development, including the differentiation of male and female flowers of the monoecious species of this species. The latter is a novelty in the control of plant sex determination.
The experiments are comprehensive and detailed. The results reported will likely be of interest to a wide audience, and the manuscript is generally well-written.
There are some problems that need to be corrected in gene expression analyses:
- Gene expression data in different tissues and abiotic stress conditions should indicate the stage of development of the plant, root, flower or fruit. It is not the same that the gene is expressed in an immature fruit than in a mature fruit. Please indicate the stages of development of all organs used.
- I assume that gene expression values are not FPKM but log2 fold change. However, this is not indicated in M&M, neither in Results, nor in Figures. It is required to include it on Figures and M&M.
- Indicate the statistical tests that you used for differential expression analysis, as well as the number of duplicates.
- Also indicate the RNA sample that has been used as a reference for differential gene expression analysis. This should appear both in material and methods and in the legend of the figures.
Figure 2 should show the exact names of the various protein domains that have been identified, not just the numbers.
Several typographical errors have been found. Please review the entire manuscript carefully.
- Line 134. Change modle by model
- Line 158. Mega 10 instead of mege 10
- Line 184. Matrix instead of matrinx
- Line 223. Our results are consistent
- .....
Author Response
Dear Reviewer,
Thank you for your recognition of our work and for your constructive suggestions. Your suggestions have greatly helped us to improve the quality of our manuscript. We have revised the manuscript following your suggestions. Please see the attachment for the point-by-point response.

Reviewer 2 Report
Liu et al. performed a phylogenetic analysis of LBD genes by using Jatropha curcas as the main object of study. As described by the authors, it provided a solid foundation for further investigation of the role of LBD genes in sexual differentiation of J. curcas. I can approve this paper for publishing after minor revision. Also, I suggested a few comment in your manuscript, please edited these comments.
A) Major comments
1. For the analysis of miRNAs, I suggest that the authors introduce the four selected miRNAs (miRNA172, miRNA156, miRNA159 and miRNA164) and point out that why these four miRNAs were selected for target prediction analysis. Also, these miRNAs should be introduced and discussed in the discussion section.
2. MCScanX software can identify multiple replication patterns of duplicated genes. Why are only fragment repeats and tandem repeats discussed in the manuscript? Please provide a revision or a reasonable explanation.
3. The authors mention more than once in the article that fragment replication events may be associated with the expansion of the LBD gene family. However, the authors did not provide a specific analysis of exactly in in and relative to which species the LBD gene family has expanded. The authors should first explore whether the LBD gene family expanded before exploring the replication patterns associated with it, or else suggest removing the relevant content.
4. In table S3, gene duplication is directed at one gene, while ka/ks is for pairs of genes, so why can they be put together? Besides, it is suggested to test the significance of ka/ks with fisher's exact test.
B) Minor comments
1. Please change cis to italic in the whole text.
2. In line 89, the “diplication” should be “duplication”.
3. In line 111, the abbreviation "A.I." should appear in full for the first time
4. In line 149, change motif2 to motif 2
5. In line 164 and 439, the 1500bp should be 1500 bp
6. What is " matrinx" in the legend of Figure 3? It should be a misspelling of the word
7. In line 343, change longger to longer
8. In line 442, change elemrnt to element
9. “Fragment duplication” or “fragmental duplication” appears several times in the text, which is not commonly called “fragmental duplication”, and it is better to unify it as “segmental duplication”.
10. In line 453, please provide the citation information for JCVI
Author Response
Dear Reviewer,
Thank you for the thoughtful and constructive comments to improve our manuscript. We have amended the manuscript following your suggestions. Point-by-point response was uploaded, please see the attachment.

Round 2
Reviewer 1 Report
In its present form the manuscript contains all the changes that have been requested. Therefore, the manuscript can be accepted for publication.